The effect of ecological characteristics on the domestication of sand rice (Agriophyllum squarrosum)

Chen Cuiyun 1 2 chency@lzb.ac.cn
Zuo Xiaoan 1 2
Zhao Xin 2
1 Key Laboratory of Ecological Safety and Sustainable Development in Arid Lands, Urat Desert-grassland Research Station, Northwest Institute of Eco-Environment and Resources, Chinese Academy of Sciences , Lanzhou , China
2 Key Laboratory of Stress Physiology and Ecology in Cold and Arid Region of Gansu Province , Lanzhou , China
Capozzi Fiore
Electronic publication date: 2024 Nov 27
Publication date: 2024
Volume: 12
Electronic Location ID: e18320
Received 2024 Jul 9; Accepted 2024 Sep 24
Copyright: © 2024 Chen et al.
Copyright year: 2024
Copyright holder: Chen et al.
License: This is an open access article distributed under the terms of the Creative Commons Attribution License, which permits unrestricted use, distribution, reproduction and adaptation in any medium and for any purpose provided that it is properly attributed. For attribution, the original author(s), title, publication source (PeerJ) and either DOI or URL of the article must be cited.
License URL: https://creativecommons.org/licenses/by/4.0/

Keywords: Sand rice, Pioneer species, Seed bank, Seed germination, Domestication

Funding: National Natural Science Foundation of China 31300226 Natural Science Youth Foundation of Gansu Province 2013GS08543 This work was supported by the National Natural Science Foundation of China [31300226] and the Natural Science Youth Foundation of Gansu Province [2013GS08543]. The funders had no role in study design, data collection and analysis, decision to publish, or preparation of the manuscript.

==============================
Sand rice (Agriophyllum squarrosum) is a pioneer species of annual plant found on mobile dunes in arid and semi-arid areas of China. Its establishment within the community could play a crucial role in the restoration of vegetation in desert environments because the ecological characteristics of sand rice make it well-suited to cope with desertification. Sand rice germinates rapidly when there is sufficient precipitation, and sand burial is beneficial for its germination. After germination, the root system rapidly extends downwards. It has short life cycles, completing the life span in 90 days at drought years. Additionally, sand rice has aerial and soil seed banks, which are suitable for arid ecosystems. Its seeds have high nutrient value of high protein quality and low carbohydrates. These ecological characteristics make sand rice a potentially environmentally friendly crop for addressing future climate change and maintaining food security, especially in desert areas. However it is unknown how ecological advantages affect the de novo domestication of sand rice. In this article, we summarize its ecological characteristics and determine optimal growth conditions for domestication and more applications in future.

Introduction

Desertification is the most important issue in drylands, affecting over 40% of the global land area (Tolba & El-Kholy, 1992; Roy et al., 2024). It leads to decreased productivity and loss of resources and poses significant environmental, economic, and social challenges for residents (Lan et al., 2014). Since 2000, global warming has further exacerbated conditions in deserts, pushing certain arid ecosystems towards becoming hyperarid ecosystems (Huang et al., 2017). Therefore, in the current scenario of climate change and limited resources, it is crucial to understand how to mitigate the degradation of vegetation ecosystems, which is a hot topic for researchers studying desertification (Li et al., 2019; Wang, Liu & Li, 2019). As ecological restoration is a critical component of food security (Abdullah et al., 2022), the driver of vegetation restoration is the community succession of pioneer species (Zhang et al., 2005). Pioneer species utilize soil water better, lower the wind velocity at the soil surface and reduce soil erosion (Li et al., 2007; Jafari et al., 2018).

Sand rice (Agriophyllum squarrosum) is the dominant pioneer species at mobile sand dune and is found in the arid and semi-arid areas of China (Chen et al., 2014; Fan et al., 2017). Its distribution area is windy, nutrient-barren, and high solar irradiance (Zhang et al., 2005; Zhao et al., 2016). Sand rice prevents nutrient loss by allometric relationships among the biomasses of different organs (Huang et al., 2009). Its lignified stems can stay in the sand for a long time and are resistant to wind erosion, which protects its small, flat seeds (Zhang et al., 2005; Lan et al., 2014). The seeds are 0.205–1.783 gram m−2, which is much higher than that of Artemisia ordosica and Caragana microphylla (Deng & Liu, 2011). Sand rice seeds are high in nutrients, containing 23.2% protein, 45.0% carbohydrates and 9.7% total fat (Zhao et al., 2021).

High-stress tolerance and nutritional value make sand rice an ideal, environmentally friendly crop of the future (Zhao et al., 2017). Climate change places increasing pressure on agriculture, destabilizing critical cropping systems with biotic or abiotic stresses (Mace et al., 2021). Developing new, tolerant varieties of crops is important for food security. In this article, we summarize the role of sand rice as a pioneer species at mobile dunes, the spatial distribution of seed banks, and its germination character, which will enlighten the current domestication of sand rice.

Survey methodology

This study comprises an integrative review based on bibliographic searches focusing on sand rice in the desertification of China conducted across the Pubmed, Elsevier and Springer databases. The review includes studies performed on topics such as “Pioneer species’’, “Seed bank”, “Germination”, or “The hindrance of ecological characteristics on the domestication of sand rice”. The main literature found in arid and semi-arid areas are organized according to our focus.

Review

Pioneer species

Sand rice has evolved a range of morphological and physiological characteristics to adapt to desert environments. It has tenacious vitality, a short life cycle, high photosynthetic efficiency, and a fast growth and reproduction rate (Li, Chang & Zhao, 1992; Zhang, Zhang & Chang, 2007). Sand rice germinates in May under sufficient precipitation conditions, grows for 2 months at nutritional stage, flowers in August, fruits in September and withers in October (Li, Chang & Zhao, 1992; Qi, An & Ye, 2010) (Fig. 1). But in drought years, the life span only lasts 90 days (Chen et al., 2014). The net photosynthetic rate, water use efficiency increase, while the transpiration rate decreases at a burial depth of 25% of seedling height (Qu et al., 2015). Its seeds germinate quickly when a precipitation event occurs and after germination, the embryonic roots rapidly develop into deep roots to resist wind erosion (Zhang et al., 2005).

Figure 1 Photos of sand rice in field and wild.

(A) Seedlings growing neatly in the field and a newly sprouted seedling. (B) Mature plants grown in the wild and the seeds.

Its resistance to cold, drought, barren, wind erosion and sand burial is proved as research continues (Li et al., 2010; Zhao et al., 2013; Qu et al., 2015). The lignified stems and withering plants slow wind speed and reduce sand dune movement (Gao et al., 2015). Sand rice has an increased growth rate, plant height and biomass compared to the control at the burial depth of one-quarter of seedling height (Zhao et al., 2013; Qu et al., 2015). When the burial depth equals the seedling height, its growth rate remains at 94.8%, and plant height and biomass are the same as in the control. Even at the burial depth of 266% of the plant height, the growth rate, plant height and biomass of sand rice are still 14.8%, 58.7%, and 42.9% of the control, respectively (Zhao et al., 2013; Li et al., 2015).

As an indicator of vegetation restoration, the community establishment of sand rice is the starting point (Zhang et al., 2003). After being settled on the mobile sand dune, sand rice grows into dense patches, and the stability of sand matrix is improved (Chang et al., 2003; Huang et al., 2009). Accompanied by the stability of sand matrix, the invasion of other species begins, and the process of community succession and restoration continues (Ma & Liu, 2008; Fan et al., 2017). With the process, species richness and diversity increase, and sand rice changes from pioneer to companion and disappears subsequently (Liu & Wang, 2009; Zhou et al., 2015).

Its role as pioneer vegetation has been proven by fencing experiments (Li et al., 2007; Liu & Wang, 2009). The biomass of sand rice accounts for 94.53% of the total biomass in Horqin Sand Land. It maintains 34.11% to the next year and 20.77% to the 3rd year, which declines to 5.08% and 0.66% at the 4th and 5th year (Zhang et al., 2003) (Fig. 2). But Zhang et al. (2005) found that in the 3rd year, sand rice still accounts for 68.08% of the total dominance in Horqin Sand Land. The dominance declines rapidly to 28.02% by year 6 and again to 0.69% by year 10 (Zhang et al., 2005). The difference could be caused by the measurement of the biomass or the total dominance.

Figure 2 The percentage of sand rice in total by fencing 1, 2, 3, 5/6 and 8/10 years.

The data of biomass came from Zhang et al. (2003); the data of dominance came from Zhang et al. (2005); the data of important value came from Li et al. (2007).

At the same time, Li et al. (2007) found that the important values of sand rice are 63.3% and 60.6% in the first and second years of fencing. The important values reduce to 41.5% in the 3rd year and to 2.1% in the 8th year (Li et al., 2007). After fencing 18 years, sand rice has completely disappeared in the sand (Zhang et al., 2005; Liu & Wang, 2009; Ma et al., 2021). In short, sand rice is dominant in the first 1–3 years of fencing. In Hunshandak Sand Land, after 1, 2, 4 and 15 years of fencing, the seed bank frequency of sand rice was 80%, 10.9%, 8.4%, and 0.6%, respectively, indicating that sand rice is dominant in the first year of fencing (Liu & Wang, 2009). Sand rice can be used to green the desert, while simultaneously providing seeds for human consumption. Its rapid growth, ability to germinate at proper burial depth, and resistance to different stresses are important characteristics for the improvement of sand rice.

Seed bank

Asynchronous seed release increases the survival of the next generation and is an effective strategy to cope with unpredictable environments (Narita & Wada, 1998; Lamont & Enright, 2000; Peters, Martorell & Ezcurra, 2011). Through asynchronous release, sand rice forms two types of seed banks (Liu et al., 2006). One is an aerial seed bank (Gutterman & Ginott, 1994; Gunster, 1994; Narita & Wada, 1998), and the other one is a soil seed bank (Thompson, 1987; Lamont, 1991).

The aerial seed bank includes canopy-stored seeds and released seeds in the air (Liu et al., 2006). Under favorable environmental conditions, the aerial seeds gradually release (Hamilton-Brown et al., 2008), with the majority dispersing from December through March (Liu et al., 2007). Notably, some seeds can remain on the plants for over a year, releasing only during strong winds during the growing season (Ma & Liu, 2008). From September to March, canopy-stored sand rice seeds significantly exceed the released aerial seeds in the Horqin Sand Land (Liu et al., 2006; Ma & Liu, 2008). However, this trend reverses after March, with a marked decline in the total aerial seed bank by May and very few remaining by July in this region (Liu et al., 2006).

In Mu Us Desert, the number of aerial seeds is 4,400 seeds m−2 in April, which is significantly more than that of soil seeds. By August, aerial seeds decrease sharply, which is 500 seeds m−2 (Gao et al., 2014). Soil seeds have a similar trend to aerial seeds in August.

The soil seed density of sand rice is the highest during first 2 months of seed maturation, then declines from November to March, and relatively stabilizes after March (Ma & Liu, 2008). Most seeds are concentrated at 0–10 cm soil depth from March to June and distribute at 10–50 cm depth after June (Ma & Liu, 2008). Soil seeds are vertically distributed at 0–70 cm depth and increase first and then decrease with increasing burial depth (Bai, Bao & Li, 2004; Liu et al., 2007). The maximum density of sand rice is 843 grains m−2 at 10–20 cm depth (Bai, Bao & Li, 2004). However, Liu et al. (2007) found that the highest seed density, 100 grains m−2, occurs at a depth of 50–60 cm. This difference might be caused by fresh seeds or persistent seeds (stored for at least 1 year). Deep buried seeds turn into shallow buried or uncovered with dune migration and/or the transition of wind erosion (Liu et al., 2007). Breaking of buried seeds helps long-distance dispersal of sand rice and leads to more extensive seed distribution (Liu & Wang, 2009).

Soil seeds are affected by different types and positions of sand dunes. Seed density reaches 40.63 grains m−2 at 0–5 cm of mobile sand dune, which is 4.3 and 6.5 times higher than that of semi-mobile and semi-fixed sand dunes (Ma et al., 2010; Hu, Ma & Gao, 2010). The seed density at 0–5 cm of fixed sand dune is the lowest. On the windward slope, the density of seeds at 0–5 cm is 152.0 grains m−2, which is more abundant than that on the leeward slope (which is 81.6 grains m−2) (Bai, Bao & Li, 2004) (Fig. 3). Seed density at 5–10 cm depth on the windward slope is 246.4 grains m−2, which is also significantly higher than on the leeward. Seed density at 5–10 cm of mobile sand dunes reaches 12.5 grains m−2, which is 4 times higher than that of semi-mobile sand dunes (Ma et al., 2010; Hu, Ma & Gao, 2010). Seeds are undetectable at 5–10 cm layer of fixed and semi-fixed sand dunes (Ma et al., 2010; Hu, Ma & Gao, 2010). At 10–20 cm and 20–30 cm depth of windward slope, seed density is 843.2 and 380.8 grains m−2 respectively, which is significantly higher than that of leeward (Fig. 3) (Bai, Bao & Li, 2004).

Figure 3 Viable seeds or plants in the persistent seed bank in different positions of the active sand dune in different arid zones.

The data of seeds m−2 in semi-arid zone came from Bai, Bao & Li (2004) and Liu et al. (2007); the data of plants m−2 in arid zone came from Ma et al. (2010).

Soil seeds germinate earlier than aerial seeds (Gao et al., 2014). Different seed germination increases the germination rate of sand rice in extreme environments (Gao et al., 2014). Germination of aerial seeds results in greater plant density than soil seeds because delayed seedling emergence helps to avoid intense competition among seedlings (Gűnster, 1994). However, soil seeds play more important ecological and evolutionary roles in plant populations (Cavieres & Arroyo, 2001; Matus, Papp & Tothmeresz, 2005). Asynchronous seeds (aerial and soil seeds) should be avoided in the domestication of sand rice. The challenge lies in preventing natural seed shedding and promoting synchronized seed maturation.

Germination

Seed germination is initiated by water imbibition at an appropriate temperature (Ma, Natalia & Abir, 2017). Sand rice germinates intermittently in April and continuously in May. The quick germination pattern in May is beneficial to achieve high population density on the mobile sand dune (Cui et al., 2007). Aerial and soil seed banks have different roles in regulating seed germination (Gao et al., 2014; Fan et al., 2017). Soil seeds germinate earlier in the growing season, while aerial seeds germinate later (Gao et al., 2014). One third of the early germinated seedlings die within 1 month due to different stresses.

Germination of sand rice is affected by environmental factors such as precipitation, temperature, dark/light, and burial depth (Tobe, Zhang & Omasa, 2005; Zheng et al., 2005; Cui et al., 2007) (Table 1). Precipitation is the most important factor in determining seed germination (Tobe, Zhang & Omasa, 2005; Liu, Zhu & Xie, 2022). Precipitation events occur mainly in the summer in the arid zone of Shapotou at the southeast edge of the Tengger Desert of Ningxia, and sand rice germinates in late spring or early summer, when a large proportion of emerging seedlings have sufficient water to survive.

Table 1 Germination characteristics of Agriophyllum squarrosum.

Sample plot	Seed collection time	Experiment time	Germination rate	Germination conditions	Optimum conditions	Dormancy or not	Authors	
Horqin Desert	Oct, 1990	Jun, 1991	17.00%	Sand bury or not before rainfall	0.2 cm sand bury	–	Li, Chang & Zhao (1992)	
Horqin Desert	May, 2001	Jun, 2001	15.20%	0–30 cm sand bury	0–5 cm depth	–	Bai, Bao & Li (2004)	
Horqin Desert	Autumn 2002	Summer 2003	94.80%	0, 2, 4, 6, 8, 10, 12 cm sand bury depth	2 cm depth	Obligatory dormancy	Li et al. (2004)	
Horqin Desert	Autumn 2003	Apr, 2004	42.10% 39.20%	Chilling or dry storage		Stronger innate dormancy	Li et al. (2006)	
Horqin Desert	Nov, 2002	May, 2003	91.30%	5/20 °C, 10/30 °C, 20/40 °C;	10/30 °C;	Light or absent	Cui et al. (2009)	
0, −0.1, −0.2, −0.4, −0.8, −1.6 MPa water potentials;	0MPa water potentials;	
0.5, 1, 2, 4, 8 cm burial depths	0.5 cm	
Horqin Desert	2004–2006	2004–2006	50.30%	Light or dark at 25 °C	Dark	–	Han (2008)	
Horqin Desert	1997, 2004–2006	1997, 2004–2006	91.20%	Light or dark at 25 °C	Dark	–	Ma et al. (2010)	
Horqin Desert	Autumn, 2003	May, 2005	35%	Chilling or dry storage		–	Ma & Liu (2008)	
Horqin Desert	Autumn, 2006	May, 2007	50%	0, 10, 20, 30, 40, 50 mm burial depths	10 mm depth;	No	Luo et al. (2009)	
30, 60, 105 mm irrigation	105 mm	
Horqin Desert	2008	2009	>90%	0.5, 2, 4 cm burial depths;	0.5 cm;	–	Zhao (2009)	
>90%	5, 10, 20 mm water supply	10 mm	
Horqin Desert	Autumn, 2007	2008	85.0%	5, 10, 15, 20, 25 mm precipitation;	25 mm;	Yes	Wang, Yan & Meng (2009)	
0%, 50%, 100% illumination;	0%;	
0.5, 1, 2, 3 cm burial depths	0.5 cm	
Mu Us Desert	Aug, 2001	May, 2002	33.30%	5/15 °C, 10/20 °C, 15/25 °C, 20/30 °C	20/30 °C
in the dark	–	Zheng et al. (2004)	
Mu Us Desert	Oct, 2011	Apr, 2012	5.00%	5/15 °C, 10/20 °C, 15/25 °C, 20/30 °C at dark	20/30 °C	Yes	Gao et al. (2014)	
Mu Us Desert	Oct, 2012	Jul, 2014	6.67%	0, 9%, 18% soil moistures at 15/25 °C;	0, 9%;	Yes	Gao et al. (2018)	
0, 1, 2, 3, 4 cycles of wetting and drying	4 cycles	
Shapotou	1993, 1994	Jul, 1996	27.40%	Sand bury or not after watering	2 cm depth	No innate dormancy	Wang, Wang & Liu (1998)	
Shapotou	2004–2006	2004–2006	23.70%	Light or dark	Dark	–	Han (2008)	
Shapotou	1997, 2004–2006	1997, 2004–2006	33.60%	Light or dark at 25 °C	Dark	–	Ma & Liu (2008)	
Shapotou	Autumn, 2008	May, 2009	99.20%	0, 1, 2, 3, 4, 6, 8 cm burial depths	1 cm	–	Fan et al. (2010)	
Minqin	2004–2006	2004–2006	100%	Light or dark	Dark	–	Han (2008)	
Minqin	1997, 2004–2006	1997, 2004–2006	95%	Light or dark at 25 °C	Dark	–	Ma & Liu (2008)	
Minqin	Autumn 2010	2011	88%	0, 2, 4, 6, 12, 18, 28, 36 g/ kg NaCl;	18 g/kg threshold;	Yes	Chen, Ma & Wang (2012)	
0, 5%, 10%, 15%, 20%, 30% PEG6000	20% threshold	
Minqin	2010	Apr, 2013	100%	0, 0.0125, 0.025, 0.05, 0.075, 0.1, 0.15 g/mL GA;	0.075, 0.1, 0.15 g/mL GA;	Yes	Wei et al. (2015)	
0, 25%, 50%, 100%, 150%, 200% NaCl;	50% NaCl;	
0, 10.6%, 16.5%, 21.3%, 25.5% PEG6000	0% PEG	
	6000	
Minqin	Autumn, 2013	2014	15%	0, 0.0125, 0.025, 0.05, 0.075, 0.1, 0.15 g/mL GA;	0.1, 0.15 g/mL GA;	Yes	Tang et al. (2017)	
0, 10.6%, 16.5%, 21.3%, 25.5% PEG6000;	0%PEG6000	
10, 15, 20, 25, 30, 35 °C;	25 °C;	
0.5, 1.5, 2.5 cm burial depths	1.5 cm depth	

The effect of temperature on seed germination is achieved by alternating temperature regimes of night and day. The germination percentage of sand rice is 93.78%, 97.82% and 71.8% at 10/20 °C, 20/30 °C and 5/15 °C (night/day) under dark conditions (Zheng et al., 2005). A higher germination percentage (96.7%) at 20/40 °C than at 10/30 °C (91.3%) and 5/20 °C (86.3%) was also found by Cui et al. (2007). However, Gao et al. (2014) found that germination percentage is 47.33%, 75.88% and not detectable at 10/20 °C, 20/30 °C, and 5/15 °C under dark conditions. The lower germination acquired by Gao et al. (2014) is due to fresh seeds compared with the persistent seeds of Zheng et al. (2005) or fresh seeds from another desert of Cui et al. (2007). No matter the difference, temperature regime of 20/30 °C is optimal for sand rice to germinate.

The germination percentage of sand rice is significantly higher in darkness than in light/dark rhythms (Gao et al., 2014). The highest germination percentage is 99.2% at 25/35 °C under dark conditions, while the lowest is 3.57% at 5/15 °C under 215 µmol m−2 s−1 light conditions (Zheng et al., 2004). In 25 μmol m−2 s−1 light conditions over a period of 2 to 12 h, germination percentage decreases from 71.30% to 11.97% at 10/20 °C (Zheng et al., 2004). In different deserts, the germination percentages of sand rice are 3.64%, 2.44% and 13.41% under 150 µmol m−2 s−1 light (Ma et al., 2021). These results indicate that light and/or low temperatures inhibit seed germination.

Seed burial depth is another important factor influencing sand rice germination. Zheng et al. (2005) report 100%, 54.8% and 2.4% germination percentages at burial depths of 0.5, 1, and 4 cm, respectively. Similarly, Cui et al. (2007) observed germination rates of 91.2%, 77.8%, and 21.6% at the same respective burial depths. The highest germination percentage is consistently achieved at a burial depth of 0.5 cm. However, Li et al. (2004) found the highest germination is 95.2% at 2 cm burial depth. This is consistent with the results of Ma et al. (2021). The optimal burial depth might be 0.5–2 cm. This suggests that a relatively shallow burial depth is preferred for efficient seed establishment and emergence.

The germination percentage of fresh sand rice seeds is relatively low, ranging from 12.2–34.1% in semi-arid zones (Deng & Liu, 2011) and 1–10% in arid zones (Liu et al., 2013). However, germination can be significantly improved through appropriate storage conditions. Zeng et al. (2014) demonstrate that storing seeds at 0–5 °C for 25 months increases the germination percentage to 93.3%. Similarly, Gao et al. (2018) found that germination increases from 28% to 100% after 1 month of storage at 30/20 °C (day/night). Dry storage of 1 month at room temperature or prolonged storage at low temperature effectively simulates seed germination of fresh seed of sand rice.

Fresh seeds germinate slowly because of dormancy. Although dormancy has the benefit of reducing the risk that germinated plants die before reproduction due to a lack of water (Wang, Wang & Liu, 1998), it limits the synchronous germination of seeds. Some research implies there is no dormancy in sand rice (Zheng et al., 2004; Tobe, Zhang & Omasa, 2005; Cui et al., 2007). However, environmental deterioration increases the probability of seed dormancy. Seeds sense environmental changes and adjust their dormancy levels to complete germination and seedling establishment (Gao et al., 2018). Sand rice has physiological dormancy (Liu et al., 2013; Ma et al., 2021), even secondary dormancy (Li et al., 2006). Dormancy break is found in seeds stored for 1 month at room temperature (Gao et al., 2018). Additionally, low temperatures, hydration-dehydration cycles, plant hormones, H2SO4 reagents, and treatment of 1.5 cm burial depth have all been proven effectively to break the dormancy of sand rice (Baskin & Baskin, 2001; Ren et al., 2019). Germination at appropriate temperature, lighting, sand burial and breaking the dormancy are necessary for sand rice improvment.

The hindrance of ecological characteristics on the domestication of sand rice

Among wild resources, sand rice has been domesticated into food crops because of its high nutritional value (Chen et al., 2014; DeHaan et al., 2016; Zhao et al., 2016). The seeds are consumed by local communities as bean jelly, and were historically used as army provision during the Tang dynasty (Zhang et al., 2018). However, its ecological characteristics, such as gradual seed release, fractional germination, and seed dormancy, have hindered breed improvement. These characteristics must be modified to obtain synchronous and stronger seedlings. Seed germination can reach almost 100% at its optimum growth conditions, including sand burial of 0.5 cm depth at 20/30 °C after fresh seeds are stored at room temperature for 1 month and then kept at 4 °C (Zheng et al., 2004; Tobe, Zhang & Omasa, 2005; Fan et al., 2017). Seed harvesting by machinery during the maturity season remains a challenge due to the issue of gradual seed release.

The yields of sand rice under natural conditions exhibit substantial variability, with reported maximum yields reaching up to 1,281 kg ha−1 (Zhang et al., 2018). Induced mutation can improve undesirable traits in sand rice genotypes, thereby accelerating the development of improved cultivars for domestication. High-yielding breeding line of GX-1 and dense planting GX-2 are currently under evaluation (Chen et al., 2014; Zhao et al., 2023). Application of delivery systems bypassing tissue culture may improve agronomic traits of sand rice (Maher et al., 2019; Cody et al., 2023). Concurrently, the implementation of efficient field management practices and the development of innovative harvesting technologies can also contribute to increased sand rice yields (Zhao et al., 2021). Collectively, these strategies ensure that sand rice and its integration as a new, climate-resilient crop can help address food security challenges in China posed by a growing population and a changing climate.

Conclusions

Based on China’s current social and natural environment, the utilization of sand rice is of great significance. Research on its ecological roles and nutritional value has attracted more and more attention. As a pioneer species, sand rice germinates quickly and in stages. The aerial and soil seed banks are efficient strategies for coping with extreme desert environments. Sometimes, seed dormancy will be induced to avoid the population disappearing under water shortage conditions. However, these characteristics are unsuitable for its domestication. We summarize its optimal growth conditions for the higher yield, including sand burial of 0.5 cm depth at 20/30 °C after fresh seeds stored at room temperature for 1 month, and then kept at 4 °C. As a potential food, sand rice can provide high nutrition to the people of China and other countries. Further funds should be invested in harvesting technology innovation, product promotion, and climate-smart agriculture practices.

Supplemental Information

Supplemental Information 1 list of references of Table 1 in English and Chinese.

We specially thank Dr. Samra for polishing the language of our manuscript.

Additional Information and Declarations

Competing Interests

Author Contributions

Data Availability

The authors declare that they have no competing interests.

Cuiyun Chen conceived and designed the experiments, performed the experiments, prepared figures and/or tables, authored or reviewed drafts of the article, and approved the final draft.

Xiaoan Zuo conceived and designed the experiments, analyzed the data, authored or reviewed drafts of the article, and approved the final draft.

Xin Zhao performed the experiments, prepared figures and/or tables, and approved the final draft.

The following information was supplied regarding data availability:

This is a literature review.

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
