# Peer review of "The effect of ecological characteristics on the domestication of sand rice (Agriophyllum squarrosum)"

_PeerJ, doi:10.7717/peerj.18320_

## Round 0.1 · original submission · Major Revisions

Please take into account the reviewers' comments and clarify in detail the novelty introduced by this article compared to the already published ones. Also, eventually check for more recent researches and add some pictures of the species.

Reviewer 1 ·

Basic reporting

Reviewers’ comments
The manuscript entitled “The effect of ecological characteristics on the domestication of sand rice (Agriophyllum squarrosum)” was interesting and impactful for sand rice researchers to further study of domestication. The content of the manuscript was good, and the author explained all sections very well. The approach and analysis of the study are sound, and the conclusions offer important new information. This research work is suitable for publication in Peer J and makes a major contribution to sand rice improvement. A few of the following minor suggestions:
1. Discuss more about de novo domestication.
2. Write 3-4 sentences synthetic conclusion (like how we can use this information to sand rice improvement) of each section-Pioneer species, Seed bank, and Germination.
3. Include some sand rice pictures.

Experimental design

Appropriate

Validity of the findings

'no comment'

Additional comments

'no comment'

Annotated reviews are not available for download in order to protect the identity of reviewers who chose to remain anonymous.

Reviewer 2 ·

Basic reporting

No comment

Experimental design

No comment

Validity of the findings

No comment

Additional comments

- In this study, the author should provide information about the life cycle of sand rice and how their seeds can be used for human consumption.

- The author should have a perspective on growing this plant as a crop. For example, the harvest time is approximately 90 days after sowing the seeds. How should the harvest be proposed?... Or should the sand rice be grown with the purpose of greening the desert, while also harvesting the seeds for human consumption?

- The author notes in the article that the citation is abbreviated as "Fig," but in the caption it is written in full. Therefore, it is necessary to adjust it to be consistent between the article and the caption.

Reviewer 3 ·

Basic reporting

I do not understand why the authors submitted another review paper, after one of the co-authors has already published a review article last year (Zhao, P. et al. (2023). Sand rice, a promising future crop for desert and marginal lands in northern China. Grassland Research, 2(4), 260–265. https://doi.org/10.1002/glr2.12065). The authors may argue that they have reviewed a different aspect of sand rice, but I still do not understand the importance. The papers cited in the MS are mostly old ones, and thus it seems many recent efforts to de novo domesticate sand rice are ignored. In addition, although this manuscript is to introduce sand rice to the community, this article does not have even a single photograph of the species. I even regret that I agreed to review this MS.

Experimental design

No commnt.

Validity of the findings

No comment.

---

## Round 0.2 · accepted · Accept

The authors have addressed the reviewers' comments.